# Association between Functional Inhibitors of Acid Sphingomyelinase (FIASMAs) and Reduced Risk of Death in COVID-19 Patients: A Retrospective Cohort Study

**DOI:** 10.3390/ph14030226

**Published:** 2021-03-07

**Authors:** Gil Darquennes, Pascal Le Corre, Olivier Le Moine, Gwenolé Loas

**Affiliations:** 1Department of Psychiatry, Hôpital Erasme, Université libre de Bruxelles (ULB), 1050 Brussels, Belgium; Gil.Darquennes@erasme.ulb.ac.be; 2Research Unit (ULB 266), Hôpital Erasme, Université libre de Bruxelles (ULB), 1050 Brussels, Belgium; 3Pôle Pharmacie, Service Hospitalo-Universitaire de Pharmacie, CHU de Rennes, 35033 Rennes, France; plcuniv@icloud.com; 4Irset (Institut de Recherche en Santé, Environnement et Travail)—UMR_S 1085, University of Rennes, CHU Rennes, INSERM, EHESP, 35000 Rennes, France; 5Laboratoire de Biopharmacie et Pharmacie Clinique, Faculté de Pharmacie, Université de Rennes 1, 35043 Rennes, France; 6Department of Medical Gastro-Enterology, Hôpital Erasme, Université libre de Bruxelles (ULB), 1050 Brussels, Belgium; Olivier.Lemoine@erasme.ulb.ac.be

**Keywords:** functional inhibitors of acid sphingomyelinase (FIASMAs), COVID-19, SARS-CoV-2, mortality, amlodipine

## Abstract

Given the current scarcity of curative treatment of COVID-19, the search for an effective treatment modality among all available medications has become a priority. This study aimed at investigating the role of functional inhibitors of acid sphingomyelinase (FIASMAs) on in-hospital COVID-19 mortality. In this retrospective cohort study, we included adult in-patients with laboratory-confirmed COVID-19 between 1 March 2020 and 31 August 2020 with definite outcomes (discharged hospital or deceased) from Erasme Hospital (Brussels, Belgium). We used univariate and multivariate logistic regression models to explore the risk factors associated with in-hospital mortality. We included 350 patients (205 males, 145 females) with a mean age of 63.24 years (SD = 17.4, range: 21–96 years). Seventy-two patients died in the hospital and 278 were discharged. The four most common comorbidities were hypertension (184, 52.6%), chronic cardiac disease (110, 31.4%), obesity (96, 27.8%) and diabetes (95, 27.1%). Ninety-three participants (26.6%) received a long-term prescription for FIASMAs. Among these, 60 (64.5%) received amlodipine. For FIASMAs status, multivariable regression showed increasing odds ratio (OR) for in-hospital deaths associated with older age (OR 1.05, 95% CI: 1.02–1.07; *p* = 0.00015), and higher prevalence of malignant neoplasm (OR 2.09, 95% CI: 1.03–4.22; *p* = 0.039). Nonsignificant decreasing OR (0.53, 95% CI: 0.27–1.04; *p* = 0.064) was reported for FIASMA status. For amlodipine status, multivariable regression revealed increasing OR of in-hospital deaths associated with older age (OR 1.04, 95% CI: 1.02–1.07; *p* = 0.0009), higher prevalence of hypertension (OR 2.78, 95% CI: 1.33–5.79; *p* = 0.0062) and higher prevalence of malignant neoplasm (OR 2.71, 95% CI: 1.23–5.97; *p* = 0.013), then secondarily decreasing OR of in-hospital death associated with long-term treatment with amlodipine (OR 0.24, 95% CI: 0.09–0.62; *p* = 0.0031). Chronic treatment with amlodipine could be significantly associated with low mortality of COVID-19 in-patients.

## 1. Introduction

The Severe Acute Respiratory Syndrome Coronavirus-2 (SARS-CoV-2) causes coronavirus disease (COVID-19). On 11 March 2020, the World Health Organization [1] declared COVID-19 as a pandemic. COVID-19 can develop into a severe illness leading to hospitalization, admission to an intensive care unit (ICU) and even death. COVID-19 has been closely related to sepsis, suggesting that most deaths on the ICU are related to viral sepsis. Based on clinical and preclinical efficacy against COVID-19, remdesivir received an emergency use authorization in the United States and Japan and was recently approved by the European Medicines Agency for the treatment of adult patients with severe COVID-19 that require supplemental oxygen [2]. However, its clinical efficacy seems to be relatively modest based on available evidence [3]. Today there are four approved COVID-19 vaccines (Oxford/AstraZeneca, Moderna, Pfizer–BioNTech and Johnson and Johnson) [4]. In the current situation of scarcity of curative treatment and pending vaccination of the entire world population, the search for an effective treatment modality among all available medications has become a priority. Acid sphingomyelinase (ASM) is a glycoprotein acting as a lysosomal hydrolase that allows for the degradation of sphingomyelin to ceramide and phosphorylcholine. The acid sphingomyelinase/ceramide system plays a crucial role in viral infection [5] and the antiviral properties of functional inhibitors of acid sphingomyelinase (FIASMAs) have been studied for several decades [6]. Several studies have reported the role of ASM in the entry of viruses (herpesvirus) [7] or the protective ability of several FIASMAs in in vivo or in vitro models (Ebola [8,9]). Moreover, several studies using murine models of sepsis have reported that a FIASMA (amitriptyline) first reduced inflammation and mortality [10], and then, improved the initial hypercoagulable state and protected septic mice from delayed coagulopathy [11]. It also protected mice from sepsis-induced brain damage through the tropomyosin receptor–kinase A signaling pathway [12]. Various drugs have been tested for their ability to inhibit ASM in vitro. Among those approved by the FDA, Kornhuber et al. [13] identified and classified 72 FIASMAs that reduce ASM activity by at least 50% at 10 µM concentration. The distribution of FIASMAs with respect to their ATC code revealed that specific therapeutic groups were over-represented: C08 (calcium channel blockers; amlodipine), D04 (antipruritics; promethazin), N04 (anti-Parkinson’s; benztropine), R06 (antihistamines for systemic use; astemizole), N06 (psychoanaleptics; fluvoxamine) and N05 (psycholeptics; chlorpromazine). Various methods have been used for the repurposing of COVID-19 drugs, such as network-based approaches, activity-based drug repositioning or in silico drug repositioning [14]. Recently, we reviewed [15] all the corresponding studies including the preprint publications and found that 32 FIASMAs could potentially be used as drugs against the SARS-CoV-2. Among these studies, one used human cells and several antidepressants including amitriptyline and fluoxetine, demonstrating an almost complete ex vivo inhibition of the infection of human epithelial cells by SARS-CoV-2 and by pp-VSV-SARS-Cov-2 spike particles [16,17]. Another study found that fluoxetine efficiently inhibited the entry and propagation of SARS-CoV-2 in a cell culture model without cytotoxic effects [18]. Moreover, three retrospective studies and one prospective study reported significant effects of FIASMAs on the prognosis of COVID-19. Retrospective studies have suggested positive effect on reducing mortality in COVID-19 hospitalized patients treated with amlodipine [19,20] or antidepressants (fluoxetine) [21]. One prospective double-blind, randomized clinical trial of fluvoxamine, one FIASMA versus placebo, found that patients (community-living nonhospitalized adults with COVID-19) treated with fluvoxamine had a low likelihood of clinical deterioration over 15 days [22]. Recently, an observational multicenter retrospective study in hospitalized COVID-19 patients reported that chlorpromazine was not associated with reduced mortality [23].

Thus, to the best of our knowledge there are 32 FIASMAs based on nonclinical or clinical studies on repurposing drugs for COVID-19. It must be noted that in almost all the studies, the FIASMA status was not identified for at least two reasons: (1) unknown by the authors or (2) other mechanisms of action explaining why the drug has been tested against SARS-CoV-2. The aim of the present study was to explore the potential role of FIASMAs in COVID-19 patients. More precisely, we tested the hypothesis, taking into account risk factors for death in patients with COVID-19 infection and receiving long-term prescriptions of FIASMAs before the infection, that these patients could have lower mortality rates than patients without FIASMAs chronic treatment.

## 2. Results

The results are shown in Table 1 and Table 2.

Ninety-three subjects received at least one FIASMA. FIASMA was prescribed as follows: 60 amlodipine, 11 amiodarone, 7 carvedilol, 5 amitriptyline, 5 desloratadine, 4 sertraline, 2 melatonine, 2 mebeverine, 2 fluoxetine, 1 hydroxyzine, 1 loperamide and 1 biperiden (see Figure 1 for the chemical structures of the twelve FIASMAs). Eighty-five patients received one FIASMA and eight received two FIASMAs. Amlodipine was prescribed for hypertension in 55 participants (92%) and for chronic cardiac disease (coronary artery disease) in five subjects. The mean prescribed daily dose of FIASMAs was 126% (SD = 64%, range: 13–400%). The frequent comorbidities were hypertension (*n* = 184, 52.6%), chronic cardiac disease (*n* = 110, 31.4%), obesity (*n*= 96, 27.8%), diabetes (*n* = 95, 27.1%), chronic kidney disease (*n* = 79, 22.6%) and chronic neurological disorders (*n* = 74, 21.1%). Seven univariate logistic regressions reported significant associations between mortality: age, hypertension, chronic cardiac disease, chronic kidney disease, chronic neurological disorders, malignant neoplasm, and dementia. The FIASMA status was not significant (Wald chi square = 0.002, *p* = 0.97). See Table 2. 

All seven significant predictors and the FIASMAs status were introduced in the multivariate logistic regression analysis. Using FIASMAs status, age and six comorbidities as predictors, the regression was significant (chi square (8) = 61.14; *p* < 0.0001). Age and malignant neoplasm were significant predictors as well. The odds ratio with 95% confidence interval (95% CI) were, respectively, 1.05 (95% CI: 1.02–1.07; *p* = 0.00015) and 2.09 (95% CI: 1.03–4.22; *p* = 0.039). For FIASMA status, there was a trend for significance (Wald chi square = 3.44, *p* = 0.064). See Table 2.

Considering that 60 (64.5%) patients received amlodipine, the association between mortality and amlodipine status was examined. In this analysis, the number of participants was not 350, but 317 (60 amlodipine and 257 FIASMAs negative). First, 17 univariate analyses were performed, and there were eight significant predictors: age, hypertension, chronic cardiac disease, chronic kidney disease, chronic neurological disorders, malignant neoplasm, chronic hematologic disease and dementia (see Table 2). Amlodipine status was not significant (Wald chi square = 2.47, *p* = 0.12). A second multivariate logistic regression was performed using amlodipine status (present, absent) as the forced variable and the eight significant predictors. The regression was significant (chi square (9) = 54.73; *p* < 0.0001). Four predictors (age, hypertension, malignant neoplasm and amlodipine) were significant. The odds ratio with 95% confidence intervals were, respectively, for age, hypertension, malignant neoplasm and amlodipine status: 1.04 (95% CI: 1.02–1.07; *p* = 0.0009), 2.78 (95% CI: 1.33–5.79; *p* = 0.0062), 2.71 (95% CI: 1.23–5.97; *p* = 0.013), 0.24 (95% CI: 0.092–0.62; *p* = 0.0031). See Table 2. Among the 184 COVID-19 patients with hypertension, 55 were treated with amlodipine (before the infection) and 129 received either antihypertensive drugs (including amlodipine prescribed after the hospitalization) or had no antihypertensive drugs. The 55 patients treated with amlodipine before the infection had a significantly lower rate of mortality (7/55, 12.7%) than the 129 patients (45/129, 34.9%) not treated with amlodipine before the infection (chi square (1) = 9.34, *p* = 0.0022).

The second most prescribed FIASMA (amiodarone) was observed in 11 participants only, and thus the potential protective effects of the other FIASMAs were not tested in the present study to take into account the validity of the logistic model.

In patients hospitalized for COVID-19 infection, age, hypertension and malignant neoplasm were significantly associated with a higher risk of mortality, whereas chronic prescription of FIASMA (notably amlodipine) was associated with a lower risk of mortality.

## 3. Discussion

The aim of this retrospective observational study was to explore the role of FIASMAs as a potential protective factor against mortality in patients hospitalized for COVID-19.

The main result was a positive association between chronic administration of FIASMAs and reduced mortality in COVID-19 in-patients. This positive association was only significant when the frequently prescribed FIASMA (amlodipine) was taken into account. Considering that 92% of the patients receiving amlodipine had hypertension, the protective effect of amlodipine was rather limited to COVID-19 in-patients with hypertension.

Several studies have tested, in vitro, the potential effect of FDA-approved drugs against SARS-CoV-2 and found that amlodipine inhibited SARS-CoV-2 replication [20,24]. In one study [20], no in vitro anti-SARS-CoV-2 effect was observed for the two other major categories of antihypertensive drugs: angiotensin-converting enzyme inhibitors and angiotensin II receptor blockers.

One retrospective study [19] on 65 elderly patients hospitalized for COVID-19 reported that 24 patients who were being treated either with amlodipine or nifedipine had a significantly lower rate of mortality (29%) than the remaining 41 patients who were not treated with calcium channel blockers (85%) (*p* = 0.0036).

Another retrospective study [20] on 96 adult COVID-19 patients with only hypertension as comorbidity, reported a significantly reduced mortality (0%) in the 19 subjects receiving amlodipine in comparison to the 77 participants who did not receive amlodipine (19.5%) (*p* = 0.037).

Amlodipine and other calcium channel blockers (CCBs) have been suggested as repurposed antiviral drugs [25] for their action on calcium homeostasis. The coronaviruses SARS-CoV and MERS-CoV use calcium ions to enable entry into the host cell via a fusion peptide derived from the spike protein. The similarity of SARS-CoV and MERS-CoV with SARS-CoV-2 suggests that a similar mechanism could be applied to the current pandemic.

Other mechanisms of potential antiviral CCB effects are anti-inflammatory and anticoagulatory effects in humans, as reported for amlodipine [26]. A third potential effect is the vasodilatory effects of CCB in the lungs and vascular system that mitigate the effects of high inflammation and hypercoagulation [19].

It is interesting to note that the inhibition of sphingomyeline was not cited by the authors [19,20,24,25,26]. Perhaps, the interest of amlodipine may be its antiviral action by its original (calcium channel blocking) and its secondary (ASM inhibition) mechanisms of action. These two mechanisms could have accumulative effects.

The present study has several limitations. First, this was a single-center, retrospective study. Second, the study focused on hospitalized patients only and this could introduce a bias in mortality and risk factors of COVID-19. Third, the relatively small sample size did not allow testing of the potential effects of the other FIASMAs. Fourth, prospective cohort studies are required to confirm results on amlodipine and to explore the potential protective role of other FIASMAs.

## 4. Materials and Methods

### 4.1. Study Design and Participants

For the present study, we used data from Erasme Hospital of Brussels (Belgium) using the International Severe Acute Respiratory and Emerging Infections Consortium (ISARIC) COVID-19 database. Data were collected and managed using REDCap (Research Electronic Data Capture), a secure, web-based software platform that allows data collection for research studies. For the present study, the database includes individuals for whom data collection commenced on or before 1 September 2020.

### 4.2. Description of the Initial Cohort

The cohort comprised 616 individuals, including 352 males and 264 females recruited from 1 March to 31 August 2020 (Appendix A). SARS-CoV-2 infection had been confirmed by laboratory testing of 354 individuals. The median age was 65 years, ranging from 19 to 96 years. Outcomes were recorded for 121 deceased patients, and the remaining follow-up is ongoing (see later). The observed mean number of days from first symptom onset to hospital admission was 11 days with a SD of 74 days and a median of 5 days. For 498 patients, complete information on the length of hospital stay was available. The observed mean number of days from hospital admission to outcome (death or discharge) was 45.3 days with a standard deviation (SD) of 67.5 days and a median of 8 days. The four most common symptoms at admission were fatigue, malaise, history of fever and shortness of breath. Two hundred and twenty-two patients received noninvasive mechanical ventilation (NIV), and 140 were admitted to the intensive care unit or high dependency unit. Amongst 394 patients with recorded outcome and details of treatment received, 40.1% received antibiotics and 48.1% received antivirals, 72.8% received some degree of oxygen supplementation, among which 54% were NIV and 11.1% received invasive mechanical ventilation.

The present study was approved by the Ethics Committee of the Erasme Hospital (Protocol P2020/358 approved on 16 July 2020). Using the initial cohort of 616 individuals, we selected 350 participants with SARS-CoV-2 infection confirmed by laboratory testing and either alive by 31 August 2020 or deceased in the hospital over the course of the hospitalization. Seventy-two patients died during hospitalization and 278 were alive. Demographic data regarding symptoms, comorbidities, laboratory findings on admission and outcomes were retrieved from the ISARIC COVID-19 database of Erasme Hospital and retrospectively reviewed and analyzed.

The Strengthening the Reporting of Cohort Studies in Epidemiology statement guidelines were followed in the conduct and reporting of the study (see online Appendix A).

Two physicians (G.D. and G.L.) were responsible for data collection. The characteristics of the sample are given in Table 1. Particular attention was given to the prescription of FIASMAs at admission. Patients with a long-term prescription of FIASMAs at admission were noted as FIASMAs positive (F+). Chronicity was defined as duration equal to or higher than seven half-life for each FIASMA (half-life range: 35–50 min (melatonin) to 20–100 days (amiodarone)). The prescribed daily dose of each FIASMA was given using the rate of defined daily dose (DDD), defined by the World Health Organization (WHO) Collaborating Centre for Drug Statistics Methodology.

In Belgium, 28 FIASMAs are available: nine psychoanaleptics—amitriptyline, clomipramine, fluoxetine, fluvoxamine, imipramine, maprotiline, nortriptyline, paroxetine, sertraline; five psycholeptics—flupentixol, hydroxyzine, melatonin, pimozide, sertindole; two antivertigo—cinnarizine, flunarizine; two antihistamines for systemic use—desloratadine, loratadine; two drugs for functional gastrointestinal disorders—alverine, mebeverine; one calcium channel blocker—amlodipine; one anticholinergic agent—biperiden; one antipropulsive—loperamide; one Class 1 and 3 antiarrhythmic drug—amiodarone; one beta blocking agent—carvedilol; one gonadotropin—clomifene; one hormone antagonist—tamoxifen; one cough suppressant—cloperastine. The participants who did not receive chronic FIASMAs on admission were noted as FIASMAs negative (F–). If a patient who did not receive chronic FIASMAs on admission had a new prescription of FIASMAs on the first day of hospitalization, then he was classified as FIASMAs negative except for melatonin (7 half-lives: 6–7 h), alverine (7 half-life: 6 h) or cloperastine (7 half-lives: 24 h).

Taking into account the prevalence of comorbidities associated with mortality in patients with COVID-19 and the disease related to the prescriptions of FIASMAs, the following comorbidities were recorded: diabetes, hypertension, chronic liver disease, chronic rheumatic disease, chronic kidney disease, chronic cardiac disease (except hypertension), malignant neoplasm, chronic neurological disorders (except dementia), dementia, chronic lung disease (except asthma), chronic hematologic disease, asthma, obesity (defined by body mass index (BMI) > 30 kg/m^2^) and smoking.

### 4.3. Statistical Analysis

Univariate and multivariate logistic regression models were used to evaluate risk factors of mortality among COVID-19 patients using Statistica (version 7.1) software [27]. The dependent variable was the status (alive or deceased) and the independent variables were sex, age, comorbidity, smoking and FIASMAs status.

From univariate analyses, each independent variable with significant association with mortality (*p* < 0.05) was included in the multivariate analysis except for the FIASMA status, which is a forced variable.

## Figures and Tables

**Figure 1 pharmaceuticals-14-00226-f001:**
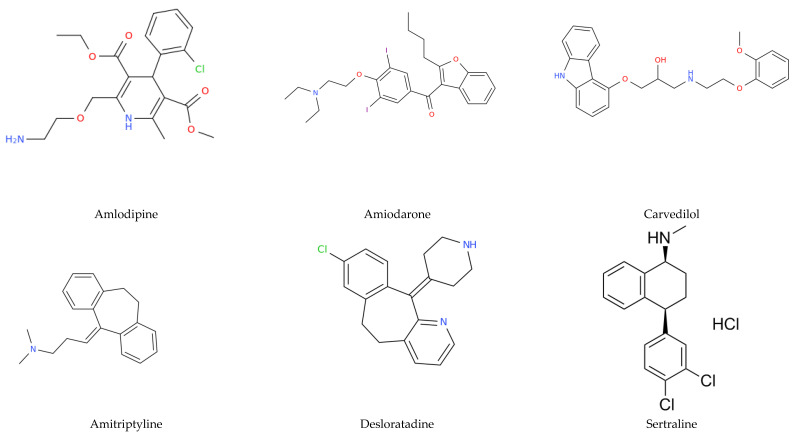
Chemical structures of the twelve FIASMAs.

**Table 1 pharmaceuticals-14-00226-t001:** Demographic and clinical characteristics of COVID-19 patients (*n* = 350).

Demographic/Clinical Characteristics.	*n (%)*
*Age*	63.24 +/− 17.4 *
*Sex*	
Female	145 (41.4)
Male	205 (58.6)
*Comorbidities*	
Hypertension	184 (52.6)
Diabetes	95 (27.1)
Chronic lung diseases	66 (18.9)
Chronic liver diseases	39 (10.3)
Chronic cardiac diseases	110 (31.4)
Chronic rheumatic disease	54 (15.4)
Chronic kidney disease	79 (22.6)
Malignant neoplasm	51 (14.6)
Chronic neurologic disorders	74 (21.1)
Dementia	28 (8)
Chronic hematologic disease	42 (12)
Asthma	28 (8)
Obesity	96 (27.8)
Smoking	39 (11.1)
*Mortality*	72 (20.6)
*FIASMAs*	93 (26.6)
*Amlodipine*	60 (17.1)

* Mean +/− SD.

**Table 2 pharmaceuticals-14-00226-t002:** Risk factors for mortality among COVID-19 patients.

*n* = 350, FIASMAs as Forced Predictor							*n* = 317, AMLODIPINE as Forced Predictor					
Variable	Univariate Analysis			Multivariable Analysis			Univariate Analysis			Multivariable Analysis		
	OR	95% CI	*p*-Value	aOR	95% CI	*p*-Value	OR	95% CI	*p*-Value	aOR	95% CI	*p*-Value
Age	1.06	1.04–1.08	<10^−4^	1.05	1.02–1.07	0.00015	1.06	1.03–1.08	<10^−4^	1.04	1.02–1.07	0.0009
Female sex (vs. male)	0.75	0.43–1.32	0.32				0.83	0.46–1.49	0.53			
Comorbidities												
Hypertension	2.88	1.63–5.08	0.0003	1.95	0.99–3.82	0.052	2.93	1.59–5.43	0.0006	2.78	1.33–5.79	0.0062
Diabetes	1.58	0.9–2.76	0.11				1.62	0.89–2.97	0.11			
Chronic lung diseases	1.6	0.86–2.97	0.14				1.51	0.76–3	0.23			
Chronic liver diseases	0.92	0.39–2.21	0.86				0.99	0.38–2.53	0.98			
Chronic cardiac diseases	2.97	1.74–5.07	<10^−4^	1.31	0.7–2.46	0.39	2.2	1.22–3.96	0.0085	0.86	0.42–1.74	0.66
Chronic rheumatic disease	1.27	0.64–2.54	0.49				1.53	0.74–3.17	0.25			
Chronic kidney disease	2.61	1.48–4.59	0.0009	1.48	0.78–2.8	0.23	2.34	1.26–4.35	0.007	1.28	0.61–2.68	0.51
Malignant neoplasm	3.06	1.62–5.8	0.0006	2.09	1.03–4.22	0.039	3.81	1.91–7.57	0.0001	2.71	1.23–5.97	0.013
Chronic neurologic disorders	2.49	1.4–4.43	0.0019	1.26	0.65–2.45	0.49	2.24	1.2–4.24	0.012	1.05	0.5–2.22	0.89
Dementia	3.27	1.47–7.3	0.0036	1.14	0.46–2.82	0.77	3.31	1.34–8.19	0.009	0.98	0.34–2.84	0.98
Chronic hematologic disease	1.65	0.8–3.43	0.17				2.22	1.04–4.72	0.038	1.15	0.47–2.83	0.76
Asthma	0.28	0.06–1.2	0.09				0.37	0.08–1.62	0.18			
Obesity	0.67	0.36–1.26	0.21				0.91	0.48–1.75	0.78			
Smoking	0.67	0.27–1.68	0.4				0.69	0.25–1.86	0.46			
FIASMAs or AMLODIPINE	0.99	0.55–1.76	0.97	0.53	0.27–1.04	0.064	0.51	0.22–1.19	0.12	0.24	0.09–0.62	0.0031

OR: odds ratio; aOR: adjusted odds ratio; 95% CI: 95% confidence interval.

## Data Availability

Anonymized individual-level data from the survey will be made available through International Severe Acute Respiratory and Emerging Infections Consortium (ISARIC) COVID-19 database. Requests for access will be reviewed by the data access committee of the Erasme Hospital.

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
