# Peer review of "Association between Functional Inhibitors of Acid Sphingomyelinase (FIASMAs) and Reduced Risk of Death in COVID-19 Patients: A Retrospective Cohort Study"

_pharmaceuticals, 2021, doi:10.3390/ph14030226_

Round 1

Reviewer 1 Report

The article in question is devoted to the study of the role of functional inhibitors of acid sphingomyelinase (FIASMA) on hospital death rate from COVID-19. Undoubtedly, the search for an effective treatment for COVID-19 is currently the most important and urgent task. This article, based on 350 patients admitted to hospital with COVID-19, describes a retrospective study of the role of FIASMA on in-hospital deaths from COVID-19. It has been shown that chronic treatment with amlodipine (one of the representatives of FIASMA) may be associated with low mortality in patients with COVID-19. I have no doubts about the validity of the applied statistical model and thus the reliability of the results. I think that the article can be published as presented. 

Author Response

Responses to reviewer 1

There is no response as the reviewer has accepted the manuscript in the original version without changes.

Reviewer 2 Report

The manuscript „Association between Functional Inhibitors of Acid Sphingomyelinase (FIASMAs) and reduced risk of death in COVID-19 patients: a retrospective cohort study by Gil Darquennes et al. describes very interesting findings on the beneficial effects antidepressants on Covid-19. The manuscript is of great interested, but several issue should be addressed.

Major problems are:

  1. The statistical power is very limited statistical power and the authors should acknowledge that they cannot explore the association of any FIASMA, but the association of amlodipine with outcomes, which is interesting (most patients with FIASMA having that medication in their sample).
  2. The case number is only meaningful for amlodipine. The authors might consider an inverse-probability-weighted analysis and then a Cox regression model.
  3. Comparing amlodipine versus not in those with hypertension would bring more compelling evidence.
  4. The authors should definitely cite the literature more carefully, also cite the several papers from Carpinteiro et al. and Hoertel et al. on the topic.
  5. The review should be rewritten to clearly reflect the data and conclusion of the manuscript.

Author Response

Responses to reviewer 2

  1. Taken into account the validity of the logistic model all the FIASMAs were grouped together using FIASMAs status variable (see page 9 line 155). Then, secondary analysis was done only on amlodipine and not for each of the other FIASMAs. This point is mentioned page 7 lines 28-30. The working hypothesis of the present study has been built taken into account the mechanism of action of FIASMAs, their bioavailability and the literature review of their action on SARS, SARS-MERS and SARS-CoV-2 (see Le Corre P, Loas G. Repurposing functional inhibitors of acid sphingomyelinase (fiasmas): an opportunity against SARS-CoV-2 infection? J Clin Pharm Ther. 2021 Mar 1). The main objective of the study was to explore the role of FIASMAs as a potential protective factor against mortality in patients hospitalized for COVID-19. All the FIASMAs were taken into account as we hypothetized a “class effect” of all inhibitors of Acid shingomyelinase. The only dependent variable was the mortality. As several FIASMAs were prescribed for comorbidities presented by COVID-19 patients as hypertension, these comorbidities were controlled in the statistical analyses. Thus a multivariate logistic regression was performed allowing to explore the potential weight of different factors (socio-demographical variables and comorbidities evidenced in univariate logistic regression). As the time between hospital admission and death or discharge was not a variable of interest in the design of the study, survival analysis and Cox regression were not used. As the study was exploratory we had a priori no idea of the prevalence of prescription of the different FIASMAs, and thus the main statistical analysis was centered on the FIASMAs status. Taken into account that Amlodipine was by far the most prescribed FIASMA, a secondary statistical analysis was done.
  2.  
  3.  See response 1
  4. As suggested by the reviewer a Chi-square analysis was done comparing in COVID-19 patients with hypertension (N = 184) the mortality rate between the patients treated by amlodipine before admission and the other subjects. The difference was significant (see page 7, lines 21-27).
  5. Four references (Carpinteiro et al (twice), Hoertel et al (twice) have been added (see references 16-17 and 21, 23) and briefly presented in the text (see page 2, Carpinteiro: lines 80-83, Hoertel: line 89 and 92-94).
  6. In the abstract, the results and the discussion we have clearly indicated that the non-planned high rate prevalence of Amlodipine prescription lead to a secondary analysis. Taken into account our results, another possibility would have been to present only the results concerning Amlodipine. However, we kept our hypothesis of research centered on all the FIASMAs taken together. 

Reviewer 3 Report

This is an interesting study on the protective effects of inhibitors of acid sphingomyelinase exerted on COVID-19 patients. The manuscript could be accepted after careful revision.

-First, english level should be seriously improved in the entire manuscript

-abstract and conclusions should be rewritten in order to make the main message clearer to the reader of Pharmaceuticals. 

-rhumatic in Table 1 and 2 should be rheumatic ?

-the authors have cited their own manuscript (ref 11) still in press on previous reports of FIASMAs and COVID-19. I would add the following references on this theme and invite them to mention their contents briefly in the text:

Schloer, Sebastian, et al. "Targeting the endolysosomal host-SARS-CoV-2 interface by clinically licensed functional inhibitors of acid sphingomyelinase (FIASMA) including the antidepressant fluoxetine." Emerging Microbes & Infections 9.1 (2020): 2245-2255.

Carpinteiro, Alexander, et al. "Pharmacological inhibition of acid sphingomyelinase prevents uptake of SARS-CoV-2 by epithelial cells." Cell Reports Medicine 1.8 (2020): 100142.

Carpinteiro, A., Edwards, M. J., Hoffmann, M., Kochs, G., Gripp, B., Weigang, S., ... & Gulbins, E. (2020). Inhibition of Acid Sphingomyelinase Blocks Infection with SARS-CoV-2. https://papers.ssrn.com/sol3/papers.cfm?abstract_id=3646562

-it is not correct to say that there is no anti-COVID-19 therapeutic approved presently. The authors should mention and cite the related reference that FDA approved remdesivir as the only specific anti-COVID-19 drug available at the moment. Of course the importance of developing other drugs remains high because remdesivir fails in some categories of patients. See and cite at least  10.1016/S1473-3099(20)30911-7 The authors should also mention that there are 3 approved vaccines against SARS-CoV-2 (citing the suitable reference(s)) and others are in advanced stages of development

-the sentence below lacks of references. Cite at least doi.org/10.3390/molecules26040986

'...Various methods are used for the repurposing of COVID-19 drugs such as network-based approaches, activity-based drug repositioning, or in silico drug repositioning.'

-I have not well understood if the protective effect on COVID-19 patients exerted by long-term administered amlodipine was excluded in case of patients with hypertension or not. Otherwise, why these patients were taking amlodipine?

-The structures in Fig. 1 are not standardized. Please use same font, size, color and provide a better quality Figure 1.

-line 100, pag 3: The mean prescribed daily dose of FIASMAs was 126%

it is percentage a way to describe dose drugs? Should not be expressed in mg(for instance)?

-lines 27-28, pag 7: rewrite the sentence 'low risk of death in participants hospitalised for COVID-19 infection an deceased at hospital'. It sounds strange to me

Author Response

Reponses to reviewer 3

  1. The manuscript was initially edited (except references and tables) by Elsevier Language Editing Services (see certificat). A new editing has been done.
  2. Changes of the abstract have been done to make clearer the main results of the study in the abstract and conclusion (see page 1, lines 32-33).
  3. Rheumatic instead of rhumatic in the text and tables.
  4. The references (Schloer et al, Carpinteiro et al (twice) have been added (see references 16-17) and briefly presented in the text (see page 2, Carpinteiro: lines 80-83 and Schloer: lines 83-85). See also response 4 to reviewer 2.
  5. Remdesivir as specific treatment anti-COVID-19 is mentioned page 2 (lines 50-54) with two references (2, 3). The 4 approved vaccines against SARS-CoV-2 were cited page 2 (lines 54-56) with one reference (4).
  6. The reference in Molecules is cited page 2 line 78 (see reference 14).
  7. See page 3, lines 113-115: Amlodipine was prescribed in 60 patients (55 for hypertension and 5 for coronary artery disease). The protective effect on chronic prescribed amlodipine is thus centered on patients with hypertension.
  8. The same font was used for the legends, and a black and white figure is now presented since we could not obtain a homogenization of the presentation (different softwares).
  9. The defined daily dose (DDD) was proposed par WHO (see page 9, lines 121-123). The mean DDD of the FIASMAs was 126% (see page 3 line 115). This parameter shows that for all the FIASMAs the daily dose was slightly higher than the daily dose recommended (i.e., x mg).
  10. The sentence page 7 lines 31-34 has been shortened.